# Homogenization of Endosymbiont Communities Hosted by Equatorial Corals during the 2016 Mass Bleaching Event

**DOI:** 10.3390/microorganisms8091370

**Published:** 2020-09-07

**Authors:** Sudhanshi S. Jain, Lutfi Afiq-Rosli, Bar Feldman, Oren Levy, Jun Wei Phua, Benjamin J. Wainwright, Danwei Huang

**Affiliations:** 1Department of Biological Sciences, National University of Singapore, 16 Science Drive 4, Singapore 117558, Singapore; lutfiafiqrosli@u.nus.edu (L.A.-R.); a0078535@u.nus.edu (J.W.P.); 2Tropical Marine Science Institute, National University of Singapore, 18 Kent Ridge Road, Singapore 119227, Singapore; 3The Mina and Everard Goodman Faculty of Life Sciences, Bar-Ilan University, Ramat Gan 5290002, Israel; barbarfel24@gmail.com (B.F.); oren.levy@biu.ac.il (O.L.); 4Yale-NUS College, National University of Singapore, 16 College Avenue West, Singapore 138527, Singapore; ben.wainwright@yale-nus.edu.sg

**Keywords:** coral reef, internal transcribed spacer, *Pachyseris speciosa*, qPCR, Scleractinia, Southeast Asia, Symbiodiniaceae

## Abstract

Thermal stress drives the bleaching of reef corals, during which the endosymbiotic relationship between Symbiodiniaceae microalgae and the host breaks down. The endosymbiont communities are known to shift in response to environmental disturbances, but how they respond within and between colonies during and following bleaching events remains unclear. In 2016, a major global-scale bleaching event hit countless tropical reefs. Here, we investigate the relative abundances of *Cladocopium* LaJeunesse & H.J.Jeong, 2018 and *Durusdinium* LaJeunesse, 2018 within and among *Pachyseris speciosa* colonies in equatorial Singapore that are known to host both these Symbiodiniaceae clades. Bleached and unbleached tissues from bleaching colonies, as well as healthy colonies, during and following the bleaching event were sampled and analyzed for comparison. The nuclear ribosomal internal transcribed spacer (ITS) regions were separately amplified and quantified using a SYBR Green-based quantitative polymerase chain reaction (qPCR) method and Illumina high-throughput sequencing. We found *Cladocopium* to be highly abundant relative to *Durusdinium*. The relative abundance of *Durusdinium*, known to be thermally tolerant, was highest in post-bleaching healthy colonies, while bleached and unbleached tissues from bleaching colonies as well as tissue from healthy colonies during the event had depressed proportions of *Durusdinium*. Given the importance of *Durusdinium* for thermal tolerance and stress response, it is surprising that bleached tissue showed limited change over healthy tissue during the bleaching event. Moreover, colonies were invariably dominated by *Cladocopium* during bleaching, but a minority of colonies were *Durusdinium*-dominant during non-bleaching times. The detailed characterization of Symbiodiniaceae in specific colonies during stress and recovery will provide insights into this crucial symbiosis, with implications for their responses during major bleaching events.

## 1. Introduction

Over half of all stony corals (Cnidaria: Anthozoa: Scleractinia) [1,2] live in obligate symbiosis with microscopic dinoflagellates of the family Symbiodiniaceae [3], which ultimately support the existence of shallow-water coral reefs in the world’s oceans [4,5]. These microalgal endosymbionts provide for nearly all of the coral hosts’ energetic requirements for growth and production of their calcium carbonate skeleton [6,7,8]. Therefore, this mutualistic relationship is one of the main foundations of the bioconstruction and functioning of coral reef ecosystems.

Symbiodiniaceae microalgae are extremely diverse, and a single coral can harbor multiple genera and species. There are at least nine recognized genus-level clades (previously known as clade A to clade I) [3], with each clade further divided into genetically diverse subclades or types [9,10,11,12]. Critically, the ability of the coral host to tolerate and recover from stress is strongly mediated by the Symbiodiniaceae community present within it, as different clades are known to have varying physiological responses to environmental conditions [13,14]. For instance, *Durusdinium* LaJeunesse 2018 (formerly clade D) is often found in corals living in regions with large temperature and turbidity fluctuations [3]. Symbiont communities, which may comprise multiple genera but are highly abundant in *Durusdinium*, could be more resistant to coral bleaching and dissociation from the host [15]. Relatedly, the abundance of *Durusdinium* relative to *Cladocopium* LaJeunesse & H.J.Jeong, 2018 (formerly clade C) is thought to be indicative of stress tolerance, with previous studies showing *Durusdinium* being dominant in coral colonies that have been affected by temperature fluctuations and bleaching events [16,17]. Other symbiotic factors of the coral holobiont, such as bacteria and viruses, are also likely to play a role in mediating coral health during environmental disturbances [18]. Indeed, the dominant Symbiodiniaceae types associated with certain host species during stress can be explained by differential abundances of particular bacterial taxa [19].

The host–symbiont relationship is influenced by many anthropogenically-driven environmental effects, such as ocean acidification, eutrophication and elevated seawater temperatures, that could threaten the health of the coral holobiont [20]. In particular, high temperatures and light levels induce thermal stress and photodamage to the coral holobiont, respectively [21]. A slight increase in temperature can cause the Symbiodiniaceae cells to be expelled from the coral host [22,23], an effect that manifests recognizably as bleaching of the live coral tissue [24]. Prolonged exceedance of seawater temperature by just 1–2 °C above the maximum monthly mean can result in severe coral bleaching and mortality [4,25,26].

Globally, coral bleaching has been occurring at higher intensities and frequencies since the 1980s [27,28,29]. From small-scale localized events, coral bleaching is now a regular and widespread phenomenon affecting major reef systems in the Caribbean [30,31,32], Red Sea [33,34,35], Indian Ocean [36,37,38,39], Indo-West Pacific [40,41,42,43,44,45], Central Pacific [46,47,48], Eastern Pacific [49,50,51] and many more. Most recently, the 2016 global-scale coral bleaching event (GCBE) hit countless tropical reefs, causing widespread damage throughout the world’s coastal ecosystems [28,52,53,54]. For example, coral reefs in Japan were severely affected, with about 40–90% coral bleaching in the Ryukyu Islands [55,56], and nearly half of all reefs in the Great Barrier Reef experienced severe (>60% of corals) bleaching [28,29]. The highly urbanized reefs in Singapore were also not spared, with bleaching recorded for up to 66% of corals in the intertidal areas and almost 60% of corals in the subtidal areas [44,57,58,59].

When bleaching is severe and corals do not recover their endosymbiont communities, a process that may involve the ‘shuffling’ (change in abundance of preexisting types) or ‘switching’ (uptake of new environmental types) of dominant and background Symbiodiniaceae types, mortality sets in [15,31,60,61]. Because corals’ susceptibility to and recovery from bleaching could be strongly influenced by the Symbiodiniaceae taxa present in the host during and after the thermal stress event, a better understanding of the diversity and ecological processes related to the symbiosis is paramount [21,31,62,63,64].

In this study, we characterize the relative abundances of *Cladocopium* and *Durusdinium* among *Pachyseris speciosa* colonies in Singapore during and after the 2016 major global bleaching event. The coral reef surveys conducted have found *P. speciosa* to be one of the most common reef corals in the turbid waters here [65,66,67]. This species was highly susceptible to bleaching during the major event in 2010 [65] and the minor event in 2013 [68], but was found to be less susceptible and better able to recover during the 2016 GCBE [44]. The locally widespread partial-colony bleaching of *P. speciosa* (Figure 1), its high propensity to recover [44] and the known association with both endosymbiont clades [10,69] allow their relative abundances to be studied among colonies of varying bleaching states. Here, we use two distinct methods—quantitative polymerase chain reaction (qPCR [12,17,70,71,72,73]) and Illumina high-throughput sequencing (HTS)—targeting the nuclear ribosomal internal transcribed spacers (ITS) [9,74,75,76] to quantify relative Symbiodiniaceae abundances. Comparing between the bleached and unbleached parts of affected colonies, unbleached colonies and post-bleaching colonies, we aim to test the relative abundances of *Durusdinium* and *Cladocopium* in the colonies sampled during and after the mass bleaching event.

## 2. Materials and Methods

### 2.1. Field Collections

In total, 39 *Pachyseris speciosa* colonies were sampled via SCUBA from Pulau Hantu (01°13′36.5″ N 103°44′47.0″ E), Singapore, during the bleaching event in July 2016. From April to July 2016, local sea surface temperatures exceeded the maximum monthly mean of 29.8 °C for several weeks, and bleaching threshold of 30.8 °C for 10 days, with a peak of 31.4 °C in May 2016 [59]. Fragments of 4–5 cm^2^ were separately collected from bleached and unbleached portions of each colony undergoing bleaching (n = 19), and from a random part of each normal, unbleached colony (n = 20). In 2017 and 2018, 3 and 23 normal, unbleached colonies respectively were sampled as post-bleaching corals. Targeted collection was done in 2018 but we included 2017 samples where available, accounting for the different numbers of samples; the two batches, both from recovered healthy corals, were lumped for analysis where necessary. Samples from 2017 were collected from Pulau Hantu, Raffles Lighthouse (01°09′36.5″ N 103°44′24.0″ E) and Kusu Island (01°13′33.0″ N 103°51′34.0″ E), whereas in 2018 they were collected from Pulau Hantu, the site sampled during the 2016 bleaching event. Individual colonies were not tagged and tracked throughout the study, but were randomly collected. All the fragments were transported to the laboratory in seawater, immediately fixed in 100% molecular grade ethanol and stored at −80 °C until processing.

### 2.2. DNA Extraction

Total genomic DNA from a 2 cm piece of tissue was extracted from each sample either with the phenol–chloroform method or DNeasy Blood and Tissue Kit (Qiagen Inc., Hilden, Germany) according to the manufacturer’s protocol. For phenol–chloroform extraction, tissue was digested overnight at 55 °C in 900 μL of CTAB (cetyltrimethylammonium bromide) buffer and 20 μL of proteinase K (20 mg/mL). The digest was treated with phenol–chloroform–isoamyl alcohol (24:25:1) to dissolve lipids and proteins and to isolate the nucleic acids. Precipitated DNA was eluted in 200 μL of water and stored at −80 °C. None of the patterns and results obtained in this study were associated with the two different DNA extraction methods.

### 2.3. Quantitative Polymerase Chain Reaction

The nuclear ribosomal internal transcribed spacer 1 (ITS1) region, located between the 18S and 5.8S rRNA genes, was targeted for the quantification of relative *Cladocopium* and *Durusdinium* rDNA copy numbers. Two samples from the collection were used to generate concentration standards for the ITS1 amplicons of *Cladocopium* and *Durusdinium*. The amplicons were quantified using Qubit dsDNA HS Assay Kit (Thermo Fisher Scientific, Waltham, MA, USA) with the Life Technologies Qubit 3.0 Fluorometer (Thermo Fisher Scientific, Waltham, MA, USA) and then diluted to 50 ng/μL. For amplification of *Cladocopium*, the universal forward primer (5′–AAG GAG AAG TCG TAA CAA GGT TTC C–3′ [70]) and *Cladocopium*-specific reverse primer (5′–AAG CAT CCC TCA CAG CCA AA–3′ [70]) were used. For *Durusdinium*, the universal forward primer and the *Durusdinium*-specific reverse primer (5′–CAC CGT AGT GGT TCA CGT GTA ATA G–3′ [70]) were used. The amplification profile consisted of an initial denaturation at 95 °C for 3 min, followed by 40 cycles of 95 °C for 30 s, 53 °C for 45 s, 72 °C for 45 s and a 5 min extension step at 72 °C. The size of the PCR products (~100 bp) was confirmed with gel electrophoresis. The products were then purified using SureClean Plus (Bioline Inc., London, UK) and quantified with the Qubit assay as mentioned above. Amplicon concentration standards were prepared by diluting the purified PCR product to 1 ng/μL, and then serially diluting to get 10 standards (1 to 10^−9^ ng/μL) each for *Cladocopium* and *Durusdinium*. All standards were run alongside each quantitative PCR (qPCR) experiment.

All DNA extracts were quantified and diluted to 1 ng/μL. SsoAdvanced SYBR Green Supermix (Bio-Rad, Hercules, CA, USA), the above-mentioned primers [70] and diluted standardized DNA template were used for qPCR using the CFX96 Touch™ Real-Time PCR Detection System (Bio-Rad, Hercules, CA, USA). All standards and samples were run in triplicates. The thermocycling profile used consists of an initial denaturation at 98 °C for 3 min followed by 40 cycles of 95 °C for 15 s and 60 °C for 30 s. A melt curve analysis from 65 °C to 95 °C (0.5 °C increase per 5 s) was subsequently performed to verify the specificity of the amplification.

CFX Manager™ Software (Bio-Rad, Hercules, CA, USA) was used to analyze the efficiency of standards and to enumerate the relative quantities of *Cladocopium* and *Durusdinium* (expressed as log-scaled ratio of *Durusdinium* concentration to *Cladocopium* concentration) in each sample based on the standard curve. Previous studies have quantified relative endosymbiont abundances by applying a correction of the ITS copy numbers based on Mieog et al. [17] (see also [70,71]). However, we here explicitly estimated the ratio of *Durusdinium* to *Cladocopium* ITS concentrations in the standardized DNA extract, which allowed comparisons among the different coral conditions and also with the relative read abundances from HTS. We note that the ratios would not reflect actual cell numbers [77].

### 2.4. Illumina Library Preparation and Sequencing

The nuclear ribosomal internal transcribed spacer 2 (ITS2) region was targeted for conventional PCR and HTS. All DNA extracts were quantified and diluted to 1 ng/μL. Reagent volumes, PCR cycling conditions and PCR clean-up protocols followed the 16S Metagenomic Sequencing Library Preparation guide for the Illumina MiSeq System [78]. This two-step protocol included an amplicon PCR (1st step) whereby the ITS2-specific SYM_VAR primer pair (SYM_VAR_5.8S2: 5′–GAATTGCAGAACTCCGTGAACC–3′ [79], SYM_VAR_REV: 5′–CGGGTTCWCTTGTYTGACTTCATGC–3′ [80]) with overhang adapters was used to generate templates for index PCRs. The index PCR (2nd step) used unique dual indexes (see Appendix A) with adapters complementary to the MiSeq flow cell for binding. Indexed PCR products were normalized using the SequalPrep^TM^ Normalization Plate Kit (Thermo Fisher Scientific, Waltham, MA, USA), pooled and sequenced under the Illumina MiSeq sequencing platform (V3 chemistry) for 300-bp paired-end reads.

Demultiplexed paired fastq.gz files containing Symbiodiniaceae ITS2 reads were analyzed using the SymPortal framework [81], run locally. Briefly, sequence filtering and a standardized quality control pipeline were conducted using mothur 1.39.5 [82], the BLAST+ suite [83] and minimum entropy decomposition (MED [84]). The community matrix output corresponding to the number of sequencing reads in each sample following the SymPortal pipeline was used for downstream Symbiodiniaceae sequence analyses.

### 2.5. Statistical Analyses

Data were analyzed using R 3.6.1 (R Core Team 2017 [85]). The ratios of *Durusdinium* to *Cladocopium* ITS1 concentrations obtained from the qPCR experiments were compared among the five groups of *Pachyseris speciosa* samples—normal (N), normal part of bleached colonies (BN), bleached part of bleached colonies (BB), post-bleaching colonies collected in 2017 (PB2017) and post-bleaching colonies collected in 2018 (PB2018). The ratios of *Durusdinium* to *Cladocopium* ITS2 read abundances were also compared among the five groups. As group variances were heterogeneous and sample sizes unequal, we used the Kruskal–Wallis test to detect differences among groups, and the post-hoc Dunn test with the Benjamini–Hochberg (1995) adjustment for multiple comparisons.

To determine if the Symbiodiniaceae communities were distinct among the five groups of *Pachyseris speciosa* samples, we used the vegan 2.5–6 package [86] to run non-metric multidimensional scaling (NMDS) and analysis of similarities (ANOSIM; 999 permutations), with the Bray–Curtis dissimilarity computed based on ITS2 subclade proportional read abundances. A Venn diagram to visualize shared and unshared subclades among the five treatments was constructed using the venn 1.9 package. To investigate Symbiodiniaceae clade dominance before, during and after the 2016 GCBE, we used a chi-square test to compare frequencies of colonies that were dominated by *Durusdinium* vs. *Cladocopium*. The test was conducted among the 15 colonies collected in July 2015 by Smith et al. [69], the 39 colonies collected in July 2016 and the 24 colonies collected in 2017 and 2018, all exclusively from the Pulau Hantu site only.

Finally, to determine if and how the qPCR tests and HTS reads were comparable in the relative *Durusdinium* vs. *Cladocopium* levels obtained, we ran a linear model to quantify the relationship between relative (*Durusdinium* divided by *Cladocopium*) ITS1 concentrations and ITS2 read abundances at the sample level. Data were log-transformed for normality and homoscedasticity.

## 3. Results

The qPCR quantification of relative concentrations of *Durusdinium* vs. *Cladocopium* ITS1 based on 84 coral samples showed significant differences among the five groups of *Pachyseris speciosa* samples (N, BN, BB, PB2017 and PB2018; Kruskal–Wallis chi-square = 29.8, *p* < 0.001; Figure 2A). Post-hoc Dunn tests indicated that the post-bleaching colonies in 2018 (PB2018) had the largest *Durusdinium* proportions, significantly greater than the unbleached (N, *p* < 0.001) and bleached (BN, *p* < 0.001; BB, *p* = 0.008) colonies in 2016. The bleached part of bleached colonies (BB) had similar proportions of *Durusdinium* compared to unbleached colonies and tissues (N and BN respectively) during the 2016 GCBE, while the post-bleaching colonies (PB2017 and PB2018) were not significantly different from each other. Overall, relative levels of *Durusdinium* vs. *Cladocopium* were more variable in corals after the bleaching event (PB2017 and PB2018; Figure 2A), with three colonies in 2018 dominated by *Durusdinium*, rather than by *Cladocopium* as was the case for all other colonies.

Illumina sequencing of the 84 coral samples yielded a total of 13,310,311 raw reads, resulting in 9,863,105 contigs. Following quality control and sequence filtering, a total of 5,250,803 Symbiodiniaceae ITS2 sequences comprising 50,897 unique sequences were retained. Across all colonies, a total of 141 Symbiodiniaceae types, comprising 61 *Cladocopium* and 80 *Durusdinium* types, were identified via the SymPortal framework. Symbiodiniaceae C27 was the most abundant type and was found in all coral samples. Symbiodiniaceae D1 was the most abundant *Durusdinium* type distributed among the normal and post-bleaching colonies, with the highest abundance in the post-bleaching colonies (Figure 3 and Figure 4). On average, each sample contained 9.9 Symbiodiniaceae types. Normal colonies from 2016 had the highest number of unique symbiont types (51), and there were only 12 types shared among all colonies (Figure 5). The richness of Symbiodiniaceae types was lower in the bleaching colonies (64) compared to normal colonies (73) during the 2016 GCBE, and this was further depressed post-bleaching (45).

The relative abundances of *Durusdinium* vs. *Cladocopium* ITS2 reads were significantly different among the five groups of *P. speciosa* samples (N, BN, BB, PB2017 and PB2018; Kruskal–Wallis chi-square = 20.6, *p* < 0.001; Figure 2B). The post-bleaching colonies in 2018 (PB2018) showed greater relative abundances of *Durusdinium* reads compared to the bleached colonies in 2016 (Dunn test; BN, *p* = 0.006; BB, *p* = 0.005). The normal colonies (N) had greater relative abundances of *Durusdinium* reads compared to both parts of bleached colonies (BN, *p* < 0.001; BB, *p* = 0.01) during the 2016 GCBE, while post-bleaching colonies (PB2017 and PB2018) were not significantly different from each other. Overall, the relative read abundances of *Durusdinium* vs. *Cladocopium* were more variable in corals after the bleaching event (PB2017 and PB2018; Figure 2B), with six colonies in 2018 dominated by *Durusdinium*, rather than by *Cladocopium* as for all other colonies (Figure 4).

The community analysis based on ITS2 proportional read abundances showed that the vast majority of the colonies during and post-GCBE had similar endosymbiont communities, forming a distinct well-populated cluster on the NMDS (Figure 6). In particular, all 20 normal colonies and nearly all of the bleached colonies during the bleaching event showed similar community structures. The six colonies dominated by *Durusdinium* in 2018 were distinct from the general cluster along the first NMDS axis. Overall, ANOSIM showed that the Symbiodiniaceae communities were distinct among the five groups of coral samples (*R* = 0.091, *p* = 0.001). Symbiodiniaceae clade dominance (*Durusdinium* vs. *Cladocopium*) in 2015, during the 2016 GCBE and in 2017–2018, were significantly differentiated based on the chi-square test for both the qPCR (*p* = 0.0454) and HTS (*p* = 0.0056) datasets.

The overall patterns of relative *Durusdinium* vs. *Cladocopium* levels were consistent between the qPCR tests and HTS reads (Figure 2). However, at the sample level, the relationship between relative ITS1 concentrations (qPCR) and ITS2 HTS read abundances was weak (*R*^2^ = 0.116, *p* = 0.00161; Appendix A). While all samples inferred to be dominated by *Cladocopium* via HTS were also inferred to be so by qPCR, three colonies from 2018 estimated to have more *Durusdinium* than *Cladocopium* by HTS were not validated by the qPCR assay.

## 4. Discussions

In this study, the relative abundances of *Cladocopium* and *Durusdinium* hosted by the common *Pachyseris speciosa* coral were compared among colonies collected during and after the 2016 GCBE in Singapore. As expected, *Cladocopium* was the most abundant Symbiodiniaceae genus in nearly all the colonies examined, though it was less diverse than *Durusdinium*, with 61 unique ITS2 sequences recovered compared to *Durusdinium*’s 80. These findings corroborate multiple characterizations of *P. speciosa*, a common and widespread coral species, including those found previously in Singapore across multiple sites [10,69] as well as in Australia [87,88,89] and Okinawa, Japan [87]. Indeed, *Cladocopium* is known to be the most abundant Symbiodiniaceae genus in many Indo-Pacific coral reef communities [3].

Based on the Illumina sequencing reads, the SymPortal framework revealed 141 unique Symbiodiniaceae ITS2 types across all colonies collected from 2016 to 2018. Among all types, clade C27 was highly abundant in all the *Pachyseris speciosa* colonies, with D1 also occurring in all samples and subclade C1 present in very low abundances in two of the colonies from 2016. On the one hand, a previous study utilizing denaturing gel gradient electrophoresis (DGGE) only found C27 as the dominant type in two of five colonies, whereas C1 and D1 were the primary types in the remaining colonies [10]. It is likely that the distinct type dominance pattern obtained in that study was due to known amplification biases affecting community diversity analyses [90]. On the other hand, Smith et al. [69] showed a dominance of the ITS2 type profile C27/C3/C3u-C115 in *Pachyseris speciosa* colonies in Pulau Hantu collected in 2015, similar to the dominant type profile of C27/C3/C27a found here. Overall, C27 was the most dominant type from 2015, through the 2016 GCBE, to 2018.

Interestingly, *Durusdinium*, which is associated with resistance to high temperatures and turbidity fluctuations [15,70,91], was found in higher relative abundance in the post-bleaching colonies from 2018. In particular, the dominance of *Cladocopium* was lost to *Durusdinium* in up to 6 of the 23 colonies in 2018, and this was also the case in 2015 among 3 of 15 *P. speciosa* colonies [69]. Because corals were either *Cladocopium* or *Durusdinium* with very high proportional abundance (>75%) and with no intermediate states, we cannot rule out the possibility that the lack of *Durusdinium*-dominant colonies during the 2016 GCBE was simply due to their mortality from bleaching. Nevertheless, even small changes in the endosymbiont community are known to be driven by various environment stressors [92,93]. The switching and/or shuffling of hosted Symbiodiniaceae clades to more tolerant and resistant species can increase survivability in adverse conditions [31,60,61]. However, the increased relative abundance and dominance of typically “background” *Durusdinium* during non-bleaching times [10,69] is surprising (but see Huang et al. [45]), since these colonies would be expected to survive during bleaching events before reverting to climax communities with more *Cladocopium* after they have recovered [94,95,96]. Our results suggest that this paradigm needs to be tested more rigorously across different coral species and environmental conditions. Indeed, endosymbiont clades hosted by *Pocillopora damicornis*, another common Indo-Pacific coral, were tracked at the colony level from bleaching through recovery during a 2014 coral bleaching event in Hawai’i, and showed limited variation in their Symbiodiniaceae profile [97].

Differences in endosymbiont community structure among the five groups of samples (N, BN, BB, PB2017 and PB2018) support the clade dominance patterns. The communities during the bleaching event were much less variable compared to those hosted by the 2018 colonies (Figure 6). Most of the latter had similar Symbiodiniaceae abundance patterns to samples collected during the 2016 GCBE, but a handful were distinct and dominated by *Durusdinium*. Evidence from previous studies has been equivocal on the relative levels of variability between stressed and unstressed corals. High temperatures have been shown to drive community differences among colonies [98,99], but they can also result in increased stability [100], as our data here illustrate. These community patterns were accompanied by a decrease in Symbiodiniaceae richness from 14.1 types per normal colony and 9.4 types per bleaching colony during the 2016 bleaching event to just 7.8 types per colony after the event (2017–2018). The contrasting and variable symbiont communities among colonies of the same species may be due to other intrinsic and extrinsic factors, including the hosts’ acclimatization capacities and spatial differences in physical conditions, respectively [101,102]. Overall, the patterns observed here underscore the flexibility of the symbiosis often associated with bleaching-susceptible corals [103], yet more research is needed to characterize and explain the basic Symbiodiniaceae richness patterns during bleaching and recovery.

qPCR has been used widely to reliably detect and quantify select taxa of Symbiodiniaceae hosted by corals [17,70,71,104,105]. While this approach has been relatively efficient and cost-effective for distinguishing Symbiodiniaceae genera, even in comparison with HTS, our method of clade quantification does not account for copy number variation and actual cell numbers [77,106]. To circumvent this limitation, we estimated the ratio of *Durusdinium* to *Cladocopium* ITS1 concentrations instead, which allowed comparisons within and among coral colonies.

For the last decade, high-throughput sequencing has been a mainstream technique for characterizing coral endosymbionts [9,75,107,108]. Most recently, development of the SymPortal analytical framework specifically for the processing of ITS2 amplicon data obtained from HTS has allowed us to characterize Symbiodiniaceae communities at a much finer resolution by accounting for intragenomic ITS2 diversity [81,109,110]. Overall, we found qPCR and HTS to produce consistent patterns of relative *Durusdinium* and *Cladocopium* abundances (Figure 2), even though the relationship between relative qPCR-estimated concentrations and HTS read abundances was weak (Appendix A). Some of these discrepancies could be due to differential sequence variabilities and amplification efficiencies between the two loci. However, the inference of clade dominance at the sample level can be considerably different, given that three colonies from 2018 found to be *Durusdinium*-dominant by HTS were *Cladocopium*-dominant based on qPCR. We thus urge caution when studying clade and type abundances, particularly for small sample sizes, and suggest verifying HTS results with qPCR for more precise abundance estimates [111,112].

While the Symbiodiniaceae community patterns are clear based on the year-on-year characterization before [69], during and after the 2016 GCBE, finer temporal sampling closer to the bleaching period would have enabled clarity regarding the hypotheses of whether the dominant clades were changing gradually or abruptly, and if particular clade-dominant colonies had higher mortality rates. The long-term, frequent and focused monitoring of corals at the colony level [113], incorporating endosymbiont genotyping, would help resolve these questions. Because there are also species-specific effects associated with bleaching, recovery and endosymbiosis in general [98,114], a broad range of species spanning different levels of susceptibility and resilience [44,115,116] needs to be studied. Corals are also associated with a complex array of bacterial communities, and it has recently been shown that the dominant Symbiodiniaceae type is linked to particular bacterial abundances under specific disturbance regimes [19]. Therefore, genotyping the different symbiotic components for an integrated analysis would enhance our understanding of the coral holobiont’s response, and enable precise predictions about their trajectories during bleaching events [19,117].

In conclusion, this study finds that *Pachyseris speciosa* colonies which had recovered from the 2016 GCBE hosted higher relative abundances of *Durusdinium* compared to colonies during the bleaching period. This pattern was driven by a minority of colonies exhibiting *Durusdinium*, rather than the *Cladocopium* dominance and stable Symbiodiniaceae community structure shown in all samples taken during the bleaching event. Together with a previous endosymbiont characterization of *P. speciosa* at the same site in 2015 [69], our findings demonstrate the homogenization of Symbiodiniaceae communities hosted by corals during the 2016 mass bleaching event. Further research is needed into the effects of bleaching on the coral–endosymbiont community and possible interactions with other factors, such as host genotype and bacterial composition. Overall, this study adds to the growing body of work on coral endosymbiont communities, and also imparts a better understanding of the contribution of various Symbiodiniaceae taxa to the resilience of urbanized reefs off Singapore.

## Figures and Tables

**Figure 1 microorganisms-08-01370-f001:**
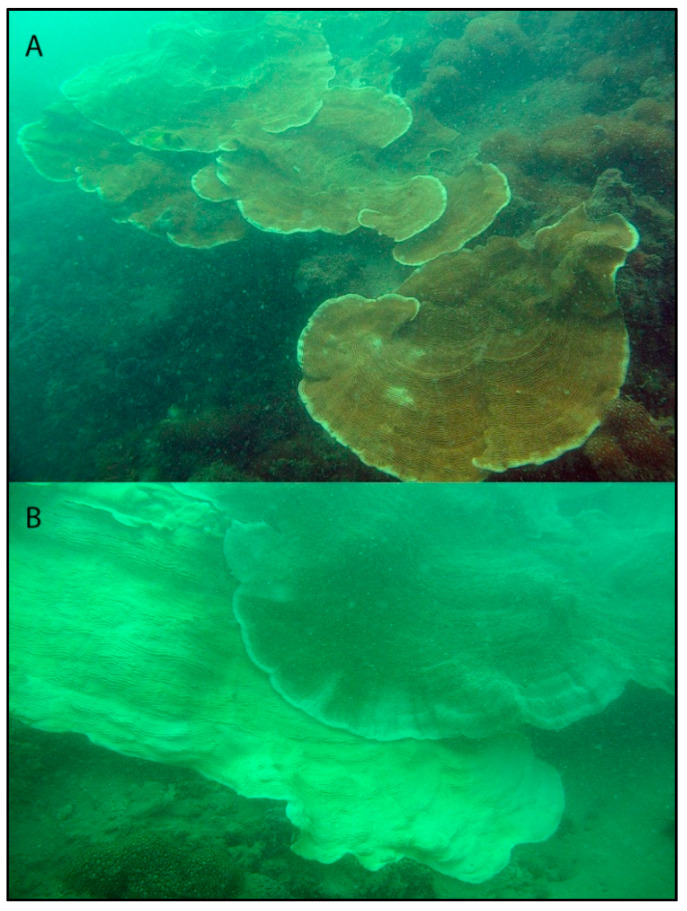
*Pachyseris speciosa* at Pulau Hantu in June 2016. (**A**) Healthy colony. (**B**) Bleached colony showing both bleached and non-bleached tissues.

**Figure 2 microorganisms-08-01370-f002:**
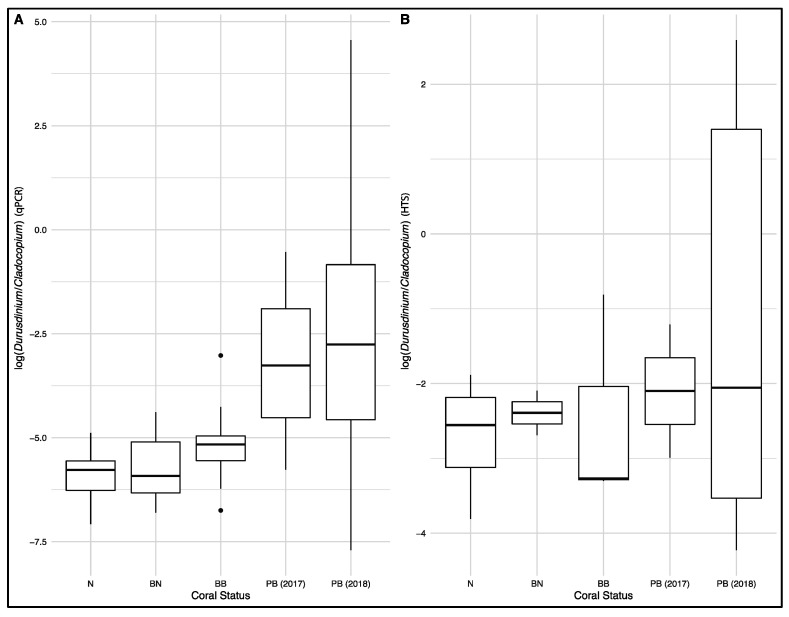
Ratios of *Durusdinium* to *Cladocopium* abundance in the five different coral groups (N: normal colonies in 2016, BN: normal portions of the bleached colonies in 2016, BB: bleached portions of the bleached colonies in 2016, PB (2017): post-bleaching colonies in 2017, PB (2018): post-bleaching colonies in 2018). (**A**) Ratios obtained from quantitative polymerase chain reaction (qPCR). (**B**) Ratios obtained from high-throughput sequencing (HTS; Illumina).

**Figure 3 microorganisms-08-01370-f003:**
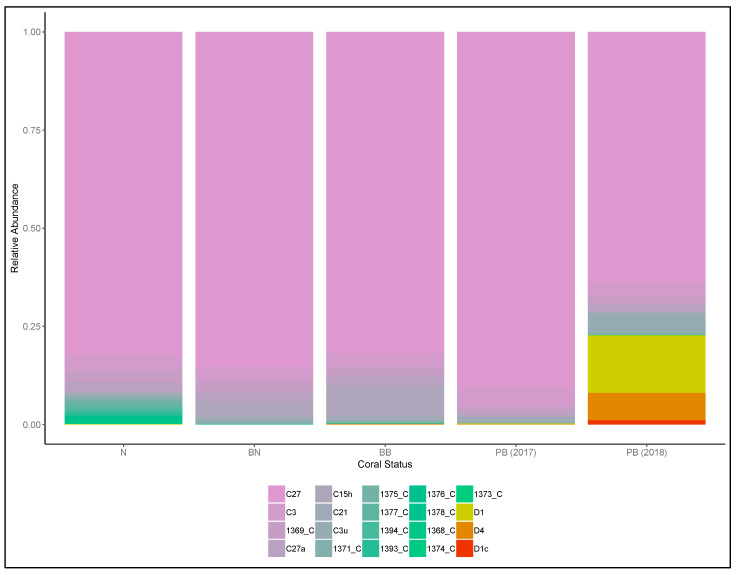
Diversity and relative abundances of Symbiodiniaceae types among the five different coral groups (N: normal colonies in 2016, BN: normal portions of the bleached colonies in 2016, BB: bleached portions of the bleached colonies in 2016, PB (2017): post-bleaching colonies in 2017, PB (2018): post-bleaching colonies in 2018).

**Figure 4 microorganisms-08-01370-f004:**
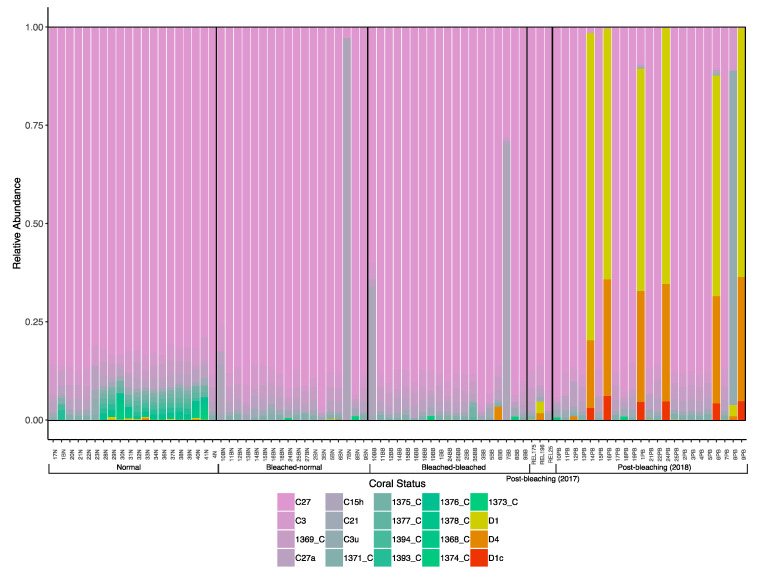
Diversity and relative abundances of Symbiodiniaceae types among all the colonies of *Pachyseris speciosa* analyzed.

**Figure 5 microorganisms-08-01370-f005:**
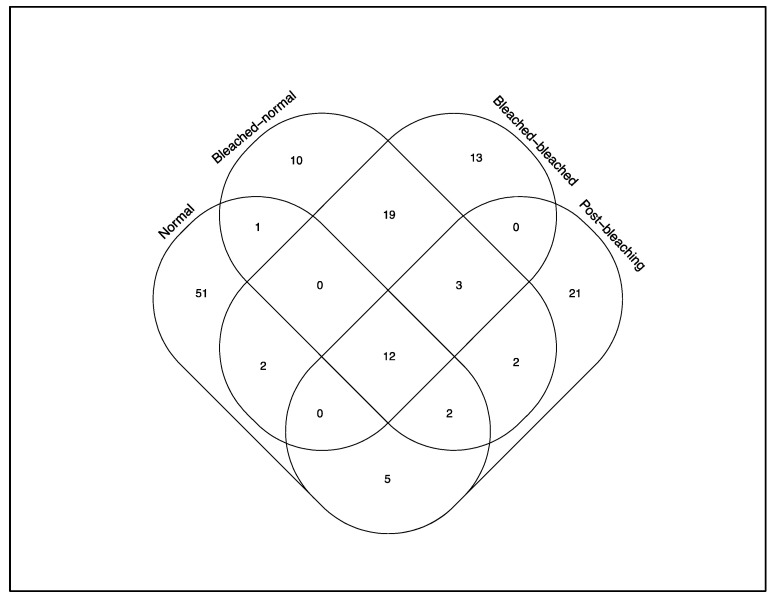
Venn diagram of shared and unique Symbiodiniaceae types among the five different coral groups. Post-bleaching corals include those from 2017 and 2018.

**Figure 6 microorganisms-08-01370-f006:**
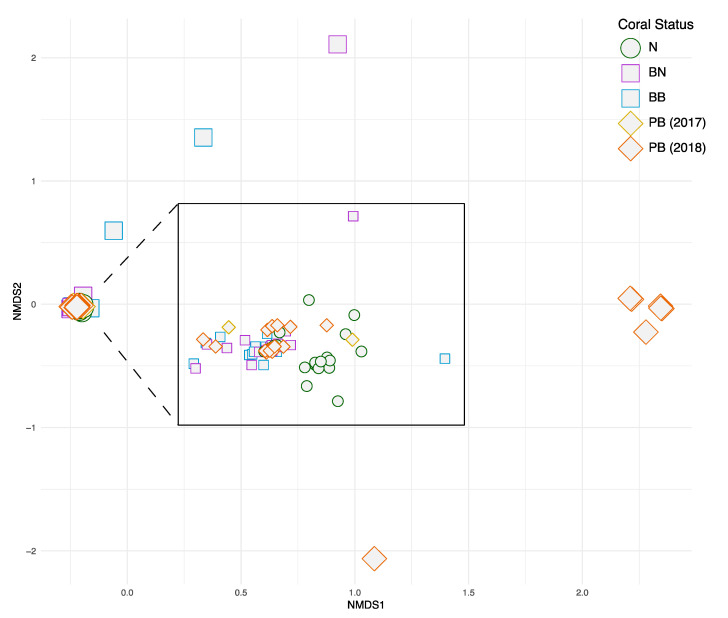
Non-metric multidimensional scaling (NMDS) based on proportional abundances of Symbiodiniaceae types with each point representing a coral sample (N: normal colonies in 2016, BN: normal portions of the bleached colonies in 2016, BB: bleached portions of the bleached colonies in 2016, PB (2017): post-bleaching colonies in 2017, PB (2018): post-bleaching colonies in 2018). Inset shows expanded main cluster.

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
