# Peer review of "Homogenization of Endosymbiont Communities Hosted by Equatorial Corals during the 2016 Mass Bleaching Event"

_microorganisms, 2020, doi:10.3390/microorganisms8091370_

Round 1

Reviewer 1 Report

This manuscript examines Symbiodiniaceae diversity prior to, during and for 2 years after a bleaching event for one common coral species. These data make a valuable contribution to an area vital for understanding the bleaching process which is arguably the most important for reef survival. Progress in this specific area has been a limited by short term experiments, so this one has a big advantage of following the corals and their symbionts for a more meaningful period. 

I feel that the research was well conducted and largely well explained. I would like some discussion on why the numbers for 2017 and 2018  samples are so different and perhaps an explanation of the resolution of the two techniques for detecting Symbiodiniaceae groups. What percentage occurrence can be detected for each sample? I apologise if this is given by reference to an article that i have missed. 

I think the manuscript is well written and the analyses makes sense. the discussion is wellconsidered and addresses limitations to techniques well. I recommend publication following minor changes. 

I have made some comments and suggested edits on the manuscript which i will attach here. 

Reviewer 2 Report

In this study, the authors investigated composition changes of symbiotic algae in reef corals during and after the mass bleaching event in 2016 that was caused by elevated seawater temperatures. They estimated relative abundance of two symbiont genera (Cladocopium and Durusdinium) in bleached and non-bleached tissues from multiple colonies. During the bleaching event, Cladocopium was abundant in all colonies, while several colonies became to be dominated by Durusdinium in 2018 after the bleaching event. This would be reasonable, because Durusdinium is known as thermal-tolerant symbionts, and similar transitions from Cladocopium to Durusdinium have been experimentally confirmed in previous studies. The manuscript was well written, and so I give only a few comments to clarify the manuscript.

Comments to the authors

1) page 4, Materials & Methods. The authors collected samples from only three colonies in 2017, and it is a quite small number compared to the other years. Please indicate the reason for this. Additionally, the authors analyzed 19 bleached colonies in 2016 and 23 post-bleaching colonies in 2018. Did you obtain tissue samples from the same individual between 2016 and 2018? If not so, please add an explanation how did you know that 23 colonies of 2018 were recovered from bleaching; I suspect that 17 of 23 colonies have not experienced a bleaching in 2016.

2) page 8, Figure 3. Cladocopium appears to be abundant (75%) in 2018 from this figure. To clearly show that 6 of 23 colonies were dominated by Durusdinium in 2018, I recommend the authors to replace this figure by supplementary Figure 1.

3) page 9, figure 5. This NMDS plot is incomprehensible, so please use more visible colors and marks.
